# Inflammation-Driven Molecular Ageing in Chronic Inflammatory Skin Diseases: Is There a Role for Biologic Therapies?

**DOI:** 10.3390/cells14181442

**Published:** 2025-09-15

**Authors:** Klara Andrzejczak, Agata Sternak, Wiktor Witkowski, Małgorzata Ponikowska

**Affiliations:** 1Faculty of Medicine, Wroclaw Medical University, Wybrzeze L. Pasteura 1, 50-367 Wroclaw, Poland; klara.andrzejczak@student.umw.edu.pl (K.A.); agata.sternak@student.umw.edu.pl (A.S.); wiktor.witkowski@student.umw.edu.pl (W.W.); 2Student Research Group of Experimental Dermatology, University Centre of General Dermatology and Oncodermatology, Wroclaw Medical University, 50-556 Wroclaw, Poland; 3Centre of General Dermatology and Oncodermatology, Wroclaw Medical University, 50-556 Wroclaw, Poland

**Keywords:** inflammageing, chronic inflammatory skin diseases, Th17 and Th2 cytokines, biologic therapy

## Abstract

Chronic inflammatory skin diseases such as atopic dermatitis, psoriasis, and hidradenitis suppurativa are systemic conditions marked by persistent immune activation. Growing evidence links them to molecular and vascular ageing, including oxidative stress, endothelial dysfunction, and reduced expression of longevity-related proteins like Klotho and SIRT1. This narrative review examines how Th17- and Th2-driven inflammation contributes to systemic inflammageing. Key cytokines—IL-17, IL-23, IL-4, IL-13, and IL-31—promote endothelial damage, oxidative stress, and metabolic dysfunction. We highlight the role of vascular biomarkers (e.g., VCAM-1, ICAM-1, ST2, P-selectin) and immune cell senescence as indicators of ageing. Finally, we explore whether biologic therapies targeting these pathways may attenuate inflammation-driven ageing. Chronic skin diseases may thus serve as accessible models of systemic inflammageing and targets for early intervention.

## 1. Introduction

Chronic inflammatory skin diseases are increasingly recognised to exert systemic effects, driven by persistent immune activation and elevated circulating cytokines rather than being confined to the skin. Among the emerging systemic consequences is molecular and vascular ageing—a process marked by endothelial dysfunction, oxidative stress, immunosenescence, and dysregulation of ageing-related genes and proteins.

Ageing impacts the body at both cellular and molecular levels, leading to impaired immune function. The gradual decline in immune reactivity, along with structural remodelling of immune organs—collectively referred to as immunosenescence—results in reduced responsiveness to vaccination and increased susceptibility to autoimmune and malignant diseases [1,2,3].

Increasing evidence indicates that immunosenescence plays a central role in a related process called inflammageing, defined as a state of low-grade, chronic inflammation that accelerates biological ageing. This process is associated with dysregulation of inflammatory molecules and impaired immune cell function, reducing the body’s ability to respond to external stimuli. Inflammageing reflects a disruption in immune system homeostasis, marked by an imbalance in pro- and anti-inflammatory signalling, and provides a framework linking immune dysregulation to age-associated decline in tissue function [1,2,4,5].

In this context, skin diseases such as psoriasis, atopic dermatitis (AD), and hidradenitis suppurativa (HS), which are now increasingly treated with targeted biologic agents, serve as clinical models to examine how chronic inflammation contributes to systemic ageing and whether therapeutic intervention may alter this trajectory. A better understanding of the mechanisms underlying these diseases and chronic inflammation has enabled the development of biologic therapies targeting cytokines and key inflammatory pathways. These therapies not only alleviate skin symptoms and improve long-term outcomes but may also modulate processes related to ageing [6,7].

This review aims to (i) summarise molecular and immunological mechanisms of ageing in chronic inflammatory skin diseases; (ii) evaluate the role of Th17 and Th2 inflammatory pathways; (iii) assess how biologic therapies may influence ageing-associated pathways; and (iv) consider implications for future research and clinical management.

## 2. Inflammageing and Molecular Ageing: Concepts and Mechanisms

### 2.1. Systemic Features of Immunosenescence

Ageing is associated with the progressive accumulation of tissue damage, driven by cellular senescence and chronic systemic inflammation, both of which play a central role in the development of many age-related diseases. This process involves not only molecular and cellular alterations but also substantial structural and functional changes in the organs responsible for immune responses and hematopoiesis. Key systemic features of immunosenescence include thymic involution and hematopoietic stem cell (HSC) dysfunction [1,3,8,9].

A key feature of immune system ageing is the accumulation of dysregulated CD4+ T lymphocytes, which are crucial for orchestrating immune responses. The thymus, one of the primary organs of the immune system, generates self-tolerant and immunocompetent T lymphocytes. Its involution, most pronounced during adolescence, gradually reduces the production of naive T lymphocytes. At the same time, it increases the proportion of terminally differentiated T cells, including memory subsets. These alterations in T lymphocyte populations and function accelerate cellular ageing, impair tissue repair, and reduce immune competence. Thymic regeneration could offer a promising approach to counteract immunosenescence [8,9,10].

Hematopoietic stem cells (HSCs) are responsible for the production of all blood cell lineages throughout life and simultaneously play a key role in the ageing of the hematopoietic system. With age, their numbers increase, but their functionality gradually declines, resulting in a loss of self-renewal and regenerative capacity [9,11,12]. Senescent HSCs tend to shift toward myelopoiesis, increasing the production of myeloid cells at the expense of normal lymphopoiesis. Prolonged exposure to chronic inflammation exacerbates HSC senescence, further impairing their function [12,13].

Immune dysregulation—including altered T cell homeostasis, a decline in stem cell function, and the accumulation of senescent cells—drives progressive tissue damage and perpetuates chronic, low-grade inflammation. This vicious cycle ultimately contributes to the development of ageing-related diseases. Understanding and modulating these mechanisms may provide new therapeutic opportunities to combat or slow immunosenescence [8,9].

### 2.2. Molecular and Endothelial Ageing

The ageing process can be examined on systemic, tissue, and molecular levels.

Molecular ageing is a complex phenomenon involving numerous changes at the cellular and molecular levels as organisms age. It involves the accumulation of damage and dysfunction within cellular structures. This includes processes such as DNA damage, changes in gene and non-coding RNA expression, genotoxicity, oxidative stress, loss of proteostasis, mitochondrial dysfunction, and the shortening of telomeres [14,15,16]. The resulting changes lead to the accumulation of damaged cells, interfering with regenerative processes and the maintenance of homeostasis, and furthermore, they become a risk factor for many diseases [15].

Endothelial ageing is the biological ageing process of endothelial cells, which form the inner lining of blood and lymphatic vessels. Endothelial cells are responsible for vascular homeostasis by maintaining blood flow and regulating vascular tone, and they also participate in proinflammatory responses and neovascularization [17,18].

During endothelial ageing, structural and functional changes occur. Endothelial cells enlarge and become flattened, and their nuclei take on a polypoid shape. Alongside these morphological changes, functional alterations are observed, such as decreased production of nitric oxide (NO), increased release of endothelin-1 (ET-1), and elevated levels of numerous proinflammatory cytokines [17,18]. These processes lead to enhanced inflammation, disruption of vascular integrity and angiogenesis, and predispose to the development of atherosclerosis [15,18].

### 2.3. Key Pathways and Phenomena: Oxidative Stress, Cellular Senescence, SIRT1, Klotho, Mitochondrial Dysfunction

#### 2.3.1. Oxidative Stress

As the immune system ages, cellular metabolism undergoes changes, leading to increased production of reactive oxygen species (ROS) [14]. ROS are generated through mitochondrial respiration, peroxisomal

β-oxidation of free fatty acids, xanthine oxidase (XO), lipoxygenase, nicotinamide adenine dinucleotide phosphate (NADPH) oxidase (NOX), microsomal cytochrome P-450 enzymes, cyclooxygenases, endothelial nitric oxide synthase (eNOS), and pro-oxidant heme molecules [18].

With advancing age, the balance between ROS production and the body’s ability to neutralise them through antioxidant mechanisms such as superoxide dismutase (SOD), catalase, and glutathione peroxidases becomes disrupted [18,19]. This imbalance is referred to as oxidative stress. ROS are molecules or atoms containing unpaired electrons, making them highly reactive and capable of damaging cellular structures such as DNA, proteins, and lipids. Moreover, they can disrupt signalling pathways, cause mitochondrial dysfunction, and contribute to inflammation and senescence [3,18,19].

#### 2.3.2. Cellular Senescence

Cellular senescence is a key component of the ageing process. At the cellular level, this process was first described by Hayflick and Moorhead, who demonstrated that human primary fibroblasts have a limited capacity to divide. This limitation results from the inability of telomeres to maintain their length after repeated replication cycles and leads to the cell entering a state of cell cycle arrest, known as replicative senescence, while remaining metabolically active [14].

Senescence can also be induced by DNA damage, oxidative stress, mitochondrial dysfunction, improper protein folding, inactivation of tumour suppressors (e.g., phosphatase and tensin homologue, PTEN) or oncogenes (e.g., Raf, BRAF), as well as radiation exposure and chemotherapy agents (Figure 1). The cellular senescence mechanism serves a protective role by helping to prevent the development of cancer through the inhibition of uncontrolled cell proliferation [14,20,21].

Senescent cells undergo numerous changes in protein expression, morphology, and metabolism. Key signalling pathways involved in these changes include those leading to the activation of the p53 and p16 INK4A [14,21].

In the early phase of cellular senescence, cells stop dividing but remain metabolically active. During this phase, the proteins p53 and p21 are essential. The p53 protein activates p21, which is an inhibitor of cyclin-dependent kinases (CDKs). p21 plays a crucial role in cell cycle regulation by causing arrest at the G1/S and G2/M checkpoints. At this stage, senescence is still reversible.

After crossing a critical point and entering an irreversible senescence state, cells progress to a subsequent phase controlled by p16 INK4a/pRb. The p16 INK4a protein inhibits CDK4 and CDK6, enzymes responsible for phosphorylating the retinoblastoma protein (pRb). Without phosphorylation, the active, hypophosphorylated form of pRb binds to E2F family transcription factors, blocking the expression of genes necessary for the transition from G1 to S phase. Consequently, the cell cannot proceed to the replication phase, resulting in permanent cell cycle arrest [21].

Senescent cells impair tissue function by altering their secretory phenotype. They produce a complex set of factors consisting of various proinflammatory cytokines, chemokines, growth factors, and matrix-remodelling enzymes, capable of modifying their microenvironment. This phenotype is known as the senescence-associated secretory phenotype (SASP) [14,15,20,21]. SASP affects neighbouring cells, inducing senescence in them. Senescent cells also undergo morphological changes such as enlargement (hypertrophy) and, in the case of adjacent cells, flattening. Senescent cell nuclei also become enlarged and so-called senescence-associated heterochromatin foci (SAHF) appear in them [14,17,18,21].

#### 2.3.3. SIRT1

Sirtuin 1 protein (SIRT1) belongs to the family of NAD^+^-dependent deacetylases and is highly conserved throughout evolution. Through the deacetylation of histone and non-histone proteins, it influences biological processes such as cellular senescence, apoptosis, carbohydrate and lipid metabolism, oxidative stress, and inflammation [22,23,24].

SIRT1 deacetylates proteins through NAD^+^ hydrolysis and the simultaneous transfer of the acetyl group from lysine residues of acetylated proteins to the 2′-OH position of ADP-ribose, ultimately producing nicotinamide and 2′-O-acetyl-ADP-ribose Through this mechanism, it regulates proteins such as p53, forkhead box transcription factors 1/3/4 (FOXO1/3/4), heat shock factor 1 (HSF1), hypoxia-inducible factor 1 alpha (HIF-1α), nuclear factor kappa B (NF-κB), p300, and TIP60. This enables SIRT1 to significantly contribute to the regulation of processes responsible for antioxidation, DNA damage repair, mitochondrial function, and autophagy [22,23,25].

##### Example Pathways

One example of a factor inhibited by SIRT1 is NF-κB. This is a key transcription factor activated in inflammatory and age-related diseases such as atherosclerosis, diabetes, and Alzheimer’s disease. SIRT1 inhibits NF-κB activity by deacetylating its p65 subunit, thereby limiting the inflammatory response. Numerous studies have demonstrated that the SIRT1–NF-κB pathway plays a significant role in counteracting cellular senescence and exerting anti-inflammatory effects [23,24,25].

Another metabolic pathway involving SIRT1 is related to adenosine 5′-monophosphate-activated protein kinase (AMPK). SIRT1 deacetylates LKB1 at Lys48, activating it and subsequently leading to increased phosphorylation of AMPK. In turn, AMPK enhances SIRT1 activity by increasing NAD^+^ levels, which promotes deacetylation of PGC-1α and activation of autophagy. This pathway represents a crucial cellular protective mechanism that supports homeostasis, promotes longevity, and counteracts senescence [23,24].

The mTOR signalling pathway also plays an important role in the regulation of autophagy. In this context, SIRT1 inhibits mTOR, thereby enhancing autophagy and reducing cellular stress [23].

Another key senescence-related pathway is the tumour suppressor pathway involving the p53 protein. Activation of p53 leads to cell cycle arrest and apoptosis. SIRT1 negatively regulates p53 by deacetylating it, thereby limiting DNA damage and reducing cellular senescence [23,24].

Moreover, SIRT1 regulates Forkhead box O (FoxO) transcription factors, which play an important role in the response to oxidative stress, DNA repair, autophagy, and cell cycle arrest. SIRT1 modulates FoxO proteins through deacetylation, enhancing their ability to induce cell cycle arrest and confer resistance to oxidative stress [23,25].

With age, the expression of SIRT1 decreases in tissues such as the liver, heart, kidneys, brain, and lungs. This decline is also observed in endothelial cells (ECs), vascular smooth muscle cells (VSMCs), and macrophages, leading to accelerated cellular senescence [23].

#### 2.3.4. Klotho

Klotho is a protein (and gene) that plays an important role in ageing processes, metabolism, and maintaining overall health. This was first observed in mice homozygous for a hypomorphic Klotho gene (kl/kl), as they exhibited shortened lifespan and stunted growth. In contrast, overexpression of this gene had the opposite effect, extending lifespan. The expression of this protein decreases with age and is further reduced in conditions such as kidney failure, diabetes, and neurodegenerative diseases like Alzheimer’s [26,27,28].

The Klotho protein has multiple functions and can exist either as a membrane-bound co-receptor for fibroblast growth factor 23 (FGF23) or as a soluble endocrine mediator with diverse biological roles. Independent of FGF23, Klotho also inhibits signalling pathways that are associated with varying degrees with the ageing process. It is involved in phosphate homeostasis regulation and exhibits antioxidant and anti-inflammatory properties. Klotho exerts protective effects in diabetes, cardiovascular diseases, and neurodegenerative disorders. It also has anti-apoptotic and anti-senescent functions, supports the maintenance of stem cells, and displays antitumor activity [26,29].

The Klotho protein plays a crucial role in modulating ageing-related processes through its influence on phosphate metabolism, fibrosis, inflammation, oxidative stress, insulin signalling, and WNT pathway activity. Phosphate homeostasis is tightly regulated by the coordinated action of the kidneys, bones, intestines, and hormones such as parathyroid hormone, vitamin D, FGF23, and Klotho. Membrane-bound and soluble forms of Klotho serve as coreceptors for FGF23, forming a complex with FGFR1c that reduces renal phosphate reabsorption and suppresses 1α-hydroxylase activity, thereby decreasing intestinal phosphate and calcium absorption through reduced vitamin D activation [26,29].

Klotho also inhibits fibrosis, particularly in the lungs and kidneys, by binding to the TGF-β type II receptor and blocking its signalling, which prevents stem cell depletion and fibrotic changes.

In terms of inflammation, Klotho suppresses the proinflammatory NF-κB pathway, reducing chronic inflammation and thereby limiting inflammageing, autoimmunity, and vascular damage [26]. Simultaneously, Klotho protects against oxidative stress by enhancing FOXO3a activity, leading to increased expression of mitochondrial antioxidant enzymes such as MnSOD [30].

Furthermore, Klotho inhibits the insulin/IGF-1 signalling pathway, which is known to promote ageing when overactive. This inhibition indirectly activates FOXO transcription factors, enhancing cellular resistance to oxidative stress [26,30].

Lastly, Klotho downregulates the WNT signalling pathway by binding to several Wnt ligands (Wnt1, Wnt3, Wnt4, Wnt5a), preventing excessive tissue remodelling, fibrosis, and organ dysfunction associated with WNT overactivation [26,30].

#### 2.3.5. Mitochondrial Dysfunction

Mitochondrial dysfunction is a hallmark of cellular senescence. It results from accumulated molecular damage, mtDNA mutations, impaired protein function, and altered membrane dynamics. Senescent cells often display enlarged mitochondria and disrupted balance between fission and fusion processes [14,15].

These changes reduce ATP production and increase reactive oxygen species (ROS) generation, contributing significantly to oxidative damage during ageing [15].

### 2.4. The Role of Chronic Inflammation and Cytokines

The clinical consequences of inflammageing are far-reaching. This persistent, low-grade inflammation in the elderly has been associated with the development and progression of various age-related chronic diseases. Inflammageing contributes not only to increased morbidity, but also to frailty, functional decline, and premature mortality. Importantly, it reflects a shift in the immune system towards a proinflammatory phenotype, even in the absence of acute infection, and often coexists with immunosenescence, further compromising immune competence [1,2,29,30].

In chronic inflammation, cytokines such as IL-1, IL-1 receptor antagonist protein (IL-1RN), IL-6, IL-8, IL-13, IL-18, C-reactive protein (CRP), interferon α (IFNα) and interferon β (IFNβ), transforming growth factor β (TGFβ), tumour necrosis factor (TNF) and its soluble receptors (members of the TNF receptor superfamily 1A and 1B), as well as serum amyloid A are present [29,30].

The production of proinflammatory cytokines is attributed to senescent cells that accumulate with age and exhibit a typical secretory phenotype called SASP [14,30]. This includes cytokines such as IL-6, IL-1, HMGB1, S100 chemokines (e.g., IL-8, MCP-1); soluble receptors (e.g., sTNFRs); metalloproteinases (e.g., collagenase); some protease inhibitors, e.g., SERPIN; and growth factors [30].

In most ageing people, chronic inflammation intensifies with age. This depends on the previously mentioned proinflammatory factors, as well as the presence of anti-inflammatory factors and the complex relationships between them [30].

Many markers are used for the clinical assessment of systemic inflammation, but C-reactive protein (CRP) and IL-6 show the strongest correlation. This is partly because IL-6 induces the production of acute phase proteins, including CRP. Moreover, IL-6 acts in an autocrine manner (on the cell that secretes it) and paracrine manner (on neighbouring cells), strengthening and maintaining the senescence state. This is a self-sustaining mechanism that causing the persistence of inflammation (Figure 2) [29,30].

Table 1 summarises the main cytokines implicated in ageing-related systemic and cellular processes. ✓ indicates documented involvement of the cytokine in the corresponding process, while empty cells indicate no reported effect in the cited references.

## 3. Biomarkers of Vascular and Molecular Ageing

Ageing is a major risk factor for vascular diseases, largely due to progressive degeneration and stiffening of the arterial wall, which ultimately compromises organ perfusion and function. A key challenge in this context is the identification of reliable biomarkers that reflect the risk and progression of vascular ageing, enabling early detection and risk stratification in susceptible individuals [40].

Biomarkers of ageing aim to predict biological age—reflecting the cumulative burden of molecular damage [41]—and often include indicators of chronic, low-grade inflammation, which plays a central role in vascular decline [40].

### 3.1. sVCAM and sICAM

sVCAM and sICAM are cell adhesion molecules that play a key role in inflammatory processes and cellular adhesion. Their expression is elevated on endothelial cells and activated leukocytes, where they mediate interactions between the endothelium and leukocytes, as well as transendothelial migration. In the early phase of inflammation, these molecules may inhibit excessive leukocyte recruitment. However, in prolonged inflammatory states, sVCAM and sICAM are released into the plasma, contributing to endothelial damage and vascular dysfunction [42,43].

A direct link has been observed between sVCAM/sICAM levels and frailty and ageing. The release of sICAM from senescent cells occurs via microparticles. Elevated concentrations of these molecules are associated with various pathological conditions, such as impaired cerebral blood flow, reduced mobility, and increased risk of falls [42]. Increased levels have also been observed in disorders such as dementia, bipolar disorder, and depression [43].

### 3.2. ST2 (Soluble Suppresion of Tumorigenity 2)

ST2 is classified within the IL-1 receptor superfamily due to its Toll/Interleukin-1 receptor (TIR) domain. It binds to IL-33, playing a key role in the inflammatory response. ST2 co-stimulates T cell activation by enhancing clonal expansion and promoting the production of cytokines and chemokines [42,44]. The binding of IL-33 to the ST2 receptor activates the nuclear transcription factor NF-κB, which regulates inflammatory responses and contributes to the pathogenesis of diseases affecting the respiratory, nervous, and urinary systems [45].

Research indicates an association between ST2 and age-related diseases, such as type 2 diabetes and cardiovascular disorders. Elevated ST2 levels have also been linked to cognitive decline [42].

### 3.3. Soluble P-Selectin

Soluble P-selectin (sP-selectin) is a soluble form of the P-selectin protein, a cell adhesion molecule. Its blood levels increase during vascular endothelial inflammation and platelet activation, making it a biomarker of both processes [46,47].

The dimeric form of soluble P-selectin binds to PSGL-1 on leukocytes, thereby mediating adhesion and signalling, and promoting inflammation. P-selectin levels are elevated in vascular inflammatory conditions such as atherosclerosis, myocardial infarction, hypertension, and venous thrombosis, which makes it a potential marker for assessing the risk of these disorders [48].

### 3.4. PDMPs

PDMPs, or Platelet-Derived Microparticles, are one of the markers of platelet activation. In the presence of arterial hypertension, platelet aggregation may be initiated, leading to the generation of PDMPs, which have prothrombotic properties. These activated particles can contribute to the development of atherosclerosis [49].

### 3.5. hsCRP

hsCRP, or high-sensitivity C-reactive protein, belongs to the pentraxin family. It plays a role in leukocyte activation, phagocytosis, and complement activation, making it a highly sensitive marker of inflammation [40,50].

Studies have shown an association between hsCRP levels and the presence of cardiovascular diseases and atherosclerosis, which is why it is considered a potential indicator or predictor of disease occurrence [50,51].

Molecular markers of longevity: Klotho, SIRT1, telomeres.

### 3.6. Klotho

Klotho deficiency leads to endothelial dysfunction, arterial stiffness, hypertension, and impaired angiogenesis, while high plasma levels reduce the risk of cardiovascular diseases, including stroke [18,52].

### 3.7. SIRT1

Studies have shown that SIRT1 expression is reduced in ageing vascular tissues, and that SIRT1 deficiency may accelerate the ageing process in endothelial cells and vascular smooth muscle cells. In contrast, activation of SIRT1 inhibits the ageing of human skin fibroblasts caused by UV radiation [23].

### 3.8. Telomeres

Telomeres are chromatin complexes composed of non-coding double-stranded tandem repeat DNA sequences and various telomere-binding proteins. These sequences are located at the ends of chromosomes and are responsible for maintaining genome stability [17]. During cell division, telomeres shorten, and their critical shortening triggers a persistent DNA damage response that leads to cellular senescence through p53-dependent mechanisms or apoptosis [18,53].

This process is a key mechanism of vascular cell ageing. Telomerase activity—the enzyme responsible for lengthening telomeres—plays a crucial role in this process. Reduced telomerase activity correlates with ageing of human endothelial cells (EC), while increased expression and activity of telomerase inhibit ageing in these cells, allowing them to maintain their function [18].

### 3.9. Measurement Possibilities and Translational Significance

Biomarkers can be valuable tools in diagnosing diseases and assessing patient condition, allowing for the selection of appropriate therapy [42]. The measurement of sVCAM-1 and sICAM-1 is performed using enzyme-linked immunosorbent assay (ELISA) on microplates. Measuring these biomarkers can be useful in assessing the risk of developing cancer, cardiovascular diseases, and type 2 diabetes [54]. The ELISA method is also used to measure soluble P-selectin [55] and hsCRP. HsCRP can be measured in both fresh and frozen plasma and is one of the most frequently analysed biomarkers in the context of cardiovascular disease risk assessment [56].

Telomere length can be assessed, for example, by the Southern blot method analysing terminal restriction fragments (TRFs). This method is applied in epidemiological studies evaluating leukocyte telomere length but can also be used to measure telomeres in other cell types [57].

Klotho and SIRT1 proteins can also be measured using the ELISA method [58]. ELISA (enzyme-linked immunosorbent assay) is a standard, rapid, and widely available diagnostic method, which enables its broad application for the assessment of various biomarkers.

## 4. Inflammatory Axes Driving Ageing in Skin Diseases

The skin, being the largest organ of the body, plays a key role as an immunological barrier, providing effective protection for the host against external factors, including pathogenic microorganisms. However, dysregulated and impaired immune responses may lead to the development of chronic inflammatory skin diseases, such as psoriasis and atopic dermatitis.

Two main immunological inflammatory pathways—the Th17 axis and the Th2 axis—play a key role in their pathogenesis, forming the inflammatory axes. Their chronic activation leads to persistent low-grade inflammation, occurring not only locally in the skin but also at the systemic level.

As previously discussed, inflammageing contributes to the acceleration of biological ageing and the progressive decline in tissue homeostasis. Its systemic effects extend beyond the immune system, negatively impacting various organs and physiological functions—including the skin, cardiovascular system, and central nervous system—thereby increasing the overall burden of age-related morbidity [59,60,61].

### 4.1. Th17 Axis

#### 4.1.1. Physiological and Immunological Roles of Th17 Cells

Th17 cells, a subset of CD4+ helper T lymphocytes, play a crucial role in the immune response to bacterial and fungal infections. They are also involved in the development of many chronic inflammatory and autoimmune diseases, such as psoriasis, hidradenitis suppurativa, alopecia areata, pityriasis rubra pilaris, pemphigus, and systemic sclerosis. They may also contribute to the pathogenesis of atopic dermatitis; however, the dominant inflammatory mechanism in this disease is the Th2 axis. Like Th1 and Th2 lymphocytes, Th17 cells are a subtype of CD4+ helper T cells. Prolonged activation of the Th17-dependent inflammatory pathway promotes inflammageing and contributes to accelerated biological ageing [62,63,64].

#### 4.1.2. Mechanisms of Th17 Lymphocyte Differentiation and Activation

The differentiation of Th17 cells requires the presence of Transforming Growth Factor β (TGF-β) and proinflammatory cytokines, especially IL-6. IL-6 activates a signalling pathway dependent on the transcription factor STAT3, which is essential for Th17 lymphocyte differentiation. Together, TGF-β and IL-6 also induce the expression of the primary Th17-specific transcription factor, retinoid-related orphan receptor-γt (ROR-γt) [65,66].

IL-23 plays a key role in maintaining and stabilising the Th17 lymphocyte phenotype and is essential for their activation. It is produced by dendritic cells (DCs), monocytes, and macrophages, which act as antigen-presenting cells (APCs) in activated skin and mucous membranes [62,67]. IL-23 is a heterodimeric cytokine composed of a unique p19 subunit and a p40 subunit shared with IL-12. It belongs to the IL-12 cytokine family, along with IL-12, IL-27, and IL-35 [68]. IL-23 is a major cytokine responsible for the survival, proliferation, and maintenance of the inflammatory response of Th17 cells [69]. Interleukin-23, acting through the IL-23R receptor—whose expression is upregulated upon activation of Th17 lymphocytes—initiates a signalling pathway involving the transcription factor STAT3 and stabilises the expression of the transcription factor ROR-γt [63].

#### 4.1.3. Pathogenic Th17 Lymphocyte Responses and the Role of Key Cytokines in the Th17 Axis

In response to stimulation by IL-23 and IL-1β, activated Th17 lymphocytes produce several proinflammatory cytokines, including IL-17A, IL-17F, IL-22, granulocyte-macrophage colony-stimulating factor (GM-CSF), and IFN-γ [31,65,66,70,71,72,73].

Among the cytokines produced by Th17 lymphocytes, IL-17 plays a particularly important role, stimulating various cell types—including endothelial cells, fibroblasts, epithelial cells, and immune cells (such as macrophages)—to produce proinflammatory cytokines (including IL-6) and chemokines like CXCL2 and CXCL8. These chemokines are involved in recruiting neutrophils to the site of inflammation. IL-17 is a key mediator of sustained chronic inflammation, especially in immune-mediated diseases such as psoriasis (Figure 3) [32,69,74,75].

Dysregulation of the IL-23/IL-17 inflammatory axis is a key mechanism in the pathogenesis of chronic inflammatory dermatoses and autoimmune diseases, leading to persistent inflammation both in the skin and systemically. This condition contributes to sustained activation of immune cells and represents one of the most important mechanisms underlying inflammageing and accelerated biological ageing [64,76].

#### 4.1.4. Mechanisms of the IL-23/IL-17 Inflammatory Axis in Vascular Ageing

Accelerated inflammatory ageing, accompanied by elevated levels of circulating proinflammatory cytokines, has profound systemic consequences. One of the most significant is vascular ageing, a key contributor to the development of cardiovascular diseases. Chronic inflammatory skin disorders—marked by sustained activation of the Th17 immune axis and excessive production of interleukin-17A (IL-17A)—may play a pivotal role in promoting this process [33,64,76,77].

Inflammageing promotes the increased production of reactive oxygen species, which damage DNA and disrupt the immune response. This damage acts as a trigger for the initiation of cellular senescence. Excessive activation of immune cells leads to elevated levels of circulating proinflammatory cytokines, sustaining inflammation and further promoting cellular senescence. As a result, cells acquire the so-called senescence-associated secretory phenotype (SASP) [76,77]. Cellular senescence is a key mechanism involved in vascular endothelial dysfunction [34].

The vascular endothelium is a single layer of cells lining blood vessels that plays a key role in regulating arterial function and maintaining vascular health. Endothelial cells synthesise and secrete a wide range of biologically active substances that act both autocrinely (on the cells that produce them) and paracrinely (on neighbouring cells). These substances are involved in vasodilation and exhibit thrombolytic and vasoprotective effects [78].

The Th17-dependent inflammatory axis, with IL-17A as a key mediator, plays a significant role in vascular ageing and the development of endotheliopathy. IL-17A induces neutrophil recruitment, increases oxidative stress, and promotes tissue remodelling. As a result, it adversely affects endothelial cells, leading to their premature senescence (endothelial cell senescence) and associated endothelial dysfunction. This results in reduced migratory capacity, impaired angiogenesis, decreased nitric oxide (NO) production, and increased vascular inflammation. These changes, collectively termed endotheliopathy, are considered a key mechanism underlying vascular ageing [18,79].

Multiple potential mechanisms may underlie the effect of IL-17A on endothelial cell senescence, one of which involves activation of the NF-κB/p53/Rb signalling pathway. In a study conducted by Liang Zhang et al., inhibition of the NF-κB pathway using ammonium pyrrolidinedithiocarbamate (PDTC) effectively suppressed IL-17A-induced expression of senescence-associated proteins [34]. Na Li et al. also demonstrated that IL-17A can promote endothelial cell senescence through activation of the p-Jun N-terminal kinase (JNK) signalling pathway, as well as regulation of FTO protein expression [79].

As a consequence, the accumulation of senescent vascular endothelial cells (VECs) increases chronic inflammation and oxidative stress, leading to progressive endothelial dysfunction and recurrent vascular damage. These processes accelerate the development and progression of atherosclerosis (AS) [80,81].

### 4.2. Th2 Axis

A classic chronic inflammatory skin disease in which the Th2 axis plays a central role in pathophysiology is atopic dermatitis (AD). The Th2 axis also contributes to the pathogenesis of other atopic diseases, including asthma, allergic rhinitis, and food allergies. The development of AD involves a complex interaction between epidermal barrier dysfunction, immunoglobulin (Ig)E sensitization, genetic predispositions, environmental factors, and immune system dysfunction [7,82]. Nevertheless, current evidence indicates that elevated IgE levels are not essential for the development of atopic dermatitis [83].

Various polymorphisms in the genes encoding IL-4, IL-13, and their receptors have been associated with a genetic predisposition to atopic dermatitis in both children and adults [80]. A study conducted by Hong Schang et al. demonstrated that the 590T and 589T alleles of the IL-4 gene may be associated with elevated serum IL-4 levels, potentially increasing the risk of atopic dermatitis development in children [84].

#### 4.2.1. The Importance and Mechanisms of Th2 Axis Activation

The Th2 axis represents a key pathway in the Th2 helper T cell-dependent immune response. Central mediators of this axis include IL-4, IL-13, and IL-31, which play crucial roles in driving inflammation in atopic dermatitis. Increased activation of this pathway is associated with eosinophilia, recruitment of basophils to inflammatory sites, and elevated IgE antibody production. These mechanisms are fundamental to the pathogenesis of the disease, contributing to the maintenance of chronic inflammation and the exacerbation of symptoms [85,86].

Epithelial cell-derived cytokines, such as IL-25, IL-33, and thymic stromal lymphopoietin (TSLP), play essential roles in initiating and amplifying the Th2-dependent immune response. They stimulate Th2 cytokine production by acting either directly on cytokine-producing cells or indirectly through effects on dendritic cells [83].

#### 4.2.2. Key Cytokines of the Th2 Axis: IL-4 and IL-13

In the context of chronic inflammation, two cytokines of the Th2 axis are of particular importance: IL-4 and IL-13. IL-4 promotes the differentiation of naive CD4+ T lymphocytes into Th2 cells, thereby initiating a Th2-dependent immune response [86]. Th2 cells secrete proinflammatory cytokines, including IL-4, IL-13, IL-5, and IL-19, which amplify the inflammatory response and recruit effector cells responsible for releasing mediators of the allergic reaction. IL-4 and IL-13 activate B lymphocytes and stimulate them to produce IgE antibodies, which bind to receptors on the surface of mast cells and basophils. This leads to their activation and the release of mediators such as histamine, contributing to the maintenance of inflammation and exacerbation of pruritus [82,87,88,89].

These cytokines induce the production of eotaxins, which promote the recruitment of additional Th2 cells and perpetuate chronic inflammation in atopic diseases [82].

The mechanism of IL-4 action begins with its binding to the IL-4Rα receptor. The resulting IL-4/IL-4Rα complex can associate with one of two receptor chains: IL-2Rγc (γc), forming the type I receptor, or IL-13Rα1, forming the type II receptor. IL-4 can signal through both receptor types, whereas IL-13 signals exclusively through the type II receptor. The receptor type depends on the cell and determines the nature of the immune response [89,90].

Upon binding to their receptor complexes, IL-4 and IL-13 activate Janus kinases (JAKs), initiating a signalling cascade that leads to the activation of signal transducer and activator of transcription 6 (STAT6), a major transcription factor [82].

IL-13 induces excessive collagen production by fibroblasts and modulates the activity of metalloproteinases MMP-9 and MMP-13, leading to increased collagen deposition and tissue remodelling. It also promotes fibrosis and the migration of inflammatory cells into the epidermis. Clinically, this manifests as thickened, lichenified skin lesions characteristic of patients with chronic atopic dermatitis [82].

#### 4.2.3. IL-31 as a Mediator of Itch and a Participant in the Neuroimmunological Axis

In addition to IL-4 and IL-13, IL-31 plays a significant role in the Th2 inflammatory axis. It belongs to the IL-6 cytokine family, and its receptor is a heterodimeric complex composed of the IL-31RA subunit and the oncostatin M receptor (OSMR) [83,91,92,93]. This complex is expressed on TRPV1+/TRPA1+ sensory neurons in the dorsal root ganglia, as well as on keratinocytes and skin immune cells, including T lymphocytes [94,95].

IL-31 is primarily produced by Th2 lymphocytes, but also by monocytes, macrophages, dendritic cells, fibroblasts, and keratinocytes [96].

The mechanism of action of IL-31 involves activation of signalling cascades through phosphorylation of the JAK/STAT and PI3K/AKT pathways, which trigger an inflammatory response [93,96].

The IL-31 neuroimmune axis plays a key role in the development of chronic itch characteristic of atopic dermatitis. Growing evidence highlights significant interactions between the immune and nervous systems in the development and persistence of chronic itch. The IL-31RA receptor is highly expressed in the dorsal root ganglia (DRGs), where the cell bodies of cutaneous sensory neurons are located [92,97]. OSMR is expressed in a subset of small-sized nociceptive neurons in the dorsal root ganglia [98].

The IL-31RA/OSMR complex on IL-31RA+/TRPV1+/TRPA1+ neurons constitutes a key neuroimmune component in the generation of Th2-dependent itch in chronic inflammatory skin diseases [37].

Itch in atopic dermatitis results from complex interactions between non-histaminergic C-type sensory nerve fibres in the skin, keratinocytes, and immune system cells [95]. IL-31, as a pruritogenic cytokine, can directly stimulate primary sensory neurons to induce itch and also modulate their sensitivity to other itch mediators such as cytokines IL-4, IL-13, TSLP, and substance P [94,99].

IL-31, as a neuroimmune modulator, also induces a distinct transcriptional programme in sensory neurons that leads to nerve fibre proliferation, which may explain why patients with atopic dermatitis exhibit increased sensitivity to minimal stimuli that provoke persistent itch [100]. The IL-31/IL-31RA immune axis may also contribute to inflammatory processes and skin barrier dysfunction, although the precise mechanisms underlying its action remain unclear [101].

Due to the pivotal role of IL-31 signalling in the pathogenesis of itch, this cytokine represents a promising therapeutic target for the treatment of chronic inflammatory skin diseases, including atopic dermatitis. One example of this approach is nemolizumab, a humanised monoclonal antibody targeting the IL-31RA receptor, which has demonstrated efficacy in alleviating itch [102,103,104].

#### 4.2.4. Epidermal Barrier Disorders Induced by Th2 Axis Cytokines

One of the key mechanisms underlying chronic inflammatory skin diseases linked to excessive activation of the Th2 axis is epidermal barrier dysfunction and dysregulation of the immune response. IL-4 and IL-13, the main cytokines of the Th2 response, impair skin barrier function, promoting the penetration of allergens and colonisation by pathogenic microorganisms such as Staphylococcus aureus. This contributes to the exacerbation of persistent skin inflammation driven by Th2 lymphocyte activity [105,106].

The skin barrier mainly consists of keratinocytes, which make up the primary population of epidermal cells, and intercellular lipids. Keratinocytes form distinct layers of the epidermis through the process of stratification. They synthesise keratins (KRTs), which are responsible for the mechanical and structural strength of the barrier. Intercellular lipids help maintain appropriate hydration levels. These components are essential for preserving the integrity of the skin barrier [35,36,107].

Filaggrin (FLG), produced by keratinocytes, plays a key role in the aggregation of keratin filaments and the formation of natural moisturising factors (NMF). Deficiency or dysfunction of filaggrin leads to increased epidermal permeability and decreased water retention capacity. Loss-of-function mutations in the FLG gene, which encodes filaggrin, are among the main risk factors for developing atopic dermatitis [89,106].

The main mediators of the Th2 response, IL-4 and IL-13, impair skin barrier function by downregulating the expression of epidermal differentiation-related molecules such as filaggrin (FLG), loricrin (LOR), and involucrin (IVL). These cytokines exert their effects by activating signalling pathways dependent on the transcription factors STAT6 and STAT3. The genes encoding these proteins—FLG, FLG2, LOR, and IVL—are located within the epidermal differentiation complex (EDC) at the chromosome 1q21.3 locus. IL-4 also inhibits the expression of genes involved in innate immunity, including those within the EDC [87,106].

Mechanisms leading to skin barrier dysfunction in chronic inflammatory skin diseases, including atopic dermatitis, may serve as a foundation for developing new and innovative therapeutic strategies [107].

#### 4.2.5. Chronic Inflammation as a Factor Accelerating Ageing in Atopic Dermatitis

Chronic activation of the Th2 inflammatory axis leads to impaired skin barrier function and promotes persistent immune activation along with cytokine-driven inflammation. This inflammation affects not only the skin but also the entire body, contributing to systemic inflammatory ageing, commonly referred to as inflammageing. Systemic inflammation can induce oxidative stress, cause endothelial dysfunction and vascular damage, and is associated with elevated levels of inflammatory biomarkers, thereby facilitating the development of comorbidities [7,60,108]

Studies have shown that chronic activation of the Th2 axis in adult patients with atopic dermatitis is associated with an increased risk of cardiovascular diseases (CVD), including coronary artery disease (CAD), and major adverse cardiovascular events (MACE). Evidence also suggests an increased incidence of venous thromboembolism (VTE). The exact mechanisms underlying the relationship between chronic systemic inflammation and elevated cardiovascular risk are not yet fully understood and require further investigation [108,109,110,111].

Proteomic studies have shown that certain markers of atherosclerosis (fractalkine/CX3CL1, CCL8, M-CSF, HGF), T cell development and activation (CD40L, IL-7, CCL25, IL-2RB, IL-15RA, CD6), and angiogenesis (VEGF-A) were significantly elevated exclusively in the serum of patients with atopic dermatitis, emphasising the systemic nature of the disease and its contribution to increased cardiovascular risk [81].

Due to the potential risk, targeted strategies for the prevention of cardiovascular complications should be considered in this patient group, especially given the high prevalence of atopic dermatitis in the population [108].

The relationship between the dominance of the Th2 axis, which promotes chronic inflammation, and systemic ageing highlights the need for further research and the development of modern, effective therapeutic strategies that modulate inflammatory axis responses and prevent the risk of systemic complications.

## 5. Biologic Therapies and Their Potential to Modulate Ageing Pathways—An Expanded Overview

### 5.1. Currently Available and Emerging Biologics

For many years, therapy for chronic inflammatory skin diseases such as atopic dermatitis, psoriasis, and hidradenitis suppurativa has been based exclusively on symptomatic treatment, aimed at alleviating symptoms through the use of topical ointments, oral antibiotics, antihistamines, corticosteroids, and retinoids. Long-term use of such therapies, especially topical and oral corticosteroids, carries a risk of adverse effects, which further underscores the need to explore alternative therapeutic strategies [6,7].

It is now established that these conditions are systemic, involving chronic and sustained activation of the immune system as well as persistent inflammation, which contributes to inflammageing and accelerated molecular ageing. Improved understanding of the pathogenesis of these dermatoses and their systemic implications has enabled the development of novel, targeted biological therapies aimed at specific cytokines and immune signalling pathways underlying the disease [6,7].

Drugs currently used to treat these conditions include tumour necrosis factor (TNF) inhibitors, selective antagonists of interleukins (IL-4, IL-17, IL-23), and the IgE inhibitor omalizumab. Increasing understanding of the mechanisms underlying chronic inflammatory diseases also permits expanding the use of available biologic drugs beyond their approved indications (off-label), which may contribute to the development of new, effective therapies. Currently, research is ongoing into drugs targeting other immune pathways, such as PGD2, JAK, Syk, and C5a inhibitors [6].

#### 5.1.1. Inhibitors of IL-4/IL-13 (Th2 Axis)

Therapies targeting the cytokines IL-4 and IL-13, key mediators of the Th2-dependent immune pathway, play a crucial role in the treatment of atopic dermatitis. This disease is characterised by a predominant immune response driven by activation of the Th2 inflammatory axis, making blockade of the key cytokines in this pathway an effective therapeutic approach [7,112,113].

Biologic drugs targeting components of the Th2 immune pathway, which are used in the treatment of atopic dermatitis, include lebrikizumab, tralokinumab, and dupilumab [112]. They demonstrate moderate effectiveness and have a favourable safety profile [114].

##### Dupilumab

Dupilumab, a monoclonal antibody targeting the interleukin-4 receptor α (IL-4Rα), blocks the signalling of both IL-4 and IL-13, key mediators of the Th2 inflammatory axis involved not only in atopic dermatitis but also in the development of other allergic diseases such as asthma [38]. It is the first biologic agent approved for the treatment of atopic dermatitis [6].

Dupilumab reduces the severity of disease symptoms and significantly improves patients’ quality of life. It also contributes to the restoration of skin lipid composition and the improvement of epidermal barrier function, which is mainly evidenced by a reduction in transepidermal water loss (TEWL). It is used in patients with moderate to severe atopic dermatitis and has a favourable safety profile, confirmed by numerous clinical trials [38,115,116,117,118,119,120,121,122]. The appropriateness of its use in continuous, long-term treatment for this patient group has been confirmed [7].

Dupilumab, in combination with standard topical therapy (e.g., topical corticosteroids, TCS), significantly improves the symptoms of atopic dermatitis in adult patients, particularly in those for whom treatment with cyclosporin A was insufficient or contraindicated. This combination is often used to increase treatment efficacy and achieve better disease control [38,123].

##### Tralokinumab

Tralokinumab is a fully human monoclonal antibody that targets interleukin-13 (IL-13), a key cytokine involved in the Th2 immune response that drives inflammation in atopic dermatitis. It binds specifically to IL-13 with high affinity, thus preventing receptor interaction and downstream signalling [124].

This drug effectively induces early and sustained improvement in clinical symptoms in patients with moderate to severe atopic dermatitis. It also exhibits an acceptable safety profile and good tolerability, including when used in combination with topical corticosteroids (TCS). The efficacy and safety of tralokinumab have been confirmed in Phase II and III clinical trials [124,125,126,127].

##### Lebrikizumab

Lebrikizumab is a promising emerging therapy for the treatment of moderate to severe atopic dermatitis, approved by the European Medicines Agency in November 2023 [128].

It is a high-affinity monoclonal antibody that selectively binds to interleukin-13 [128,129]. This drug prevents the formation of the IL-4Rα/IL-13Rα1 heterodimer receptor signalling complex, effectively blocking the IL-13-dependent signalling pathway and reducing the associated inflammatory process [130].

Clinical trials have confirmed good tolerability of lebrikizumab across various age groups. Most treatment-emergent adverse events (TEAEs) were mild or moderate in severity and did not require treatment discontinuation [129,131,132].

#### 5.1.2. Inhibitors of IL-17/IL-17F (Th17 Axis)

The latest biologic drugs approved for the treatment of moderate to severe psoriasis include inhibitors of interleukin (IL)-17 and IL-23 [133].

Therapeutic approaches targeting interleukin-17 (IL-17), a key mediator of the Th17-dependent immune pathway, play a crucial role in the treatment of many chronic inflammatory and autoimmune skin diseases, such as psoriasis and hidradenitis suppurativa. The chronic and excessive activation of the Th17-dependent IL-23/IL-17 inflammatory axis in these conditions justifies the use of effective therapies targeting the key cytokines in this pathway. IL-17 promotes the inflammatory response and disrupts immunoregulatory mechanisms, thus representing a promising therapeutic target in this group of diseases. Interleukin-17 inhibitors include secukinumab, ixekizumab, and bimekizumab [6,134,135].

Newer biologics targeting IL-17 or the p19 subunit of IL-23 have demonstrated superior efficacy, as evidenced by a higher proportion of patients achieving PASI 90 compared with earlier biologic therapies such as ustekinumab and tumour necrosis factor alpha (TNF-α) inhibitors adalimumab, certolizumab, and etanercept [135,136].

##### Secukinumab

Secukinumab is a recombinant, high-affinity, fully human IgG1κ monoclonal antibody (mAb) that selectively binds and neutralises interleukin-17A (IL-17A), thus preventing its interaction with the IL-17 receptor and downstream signalling. It is approved for the treatment of moderate to severe psoriasis, psoriatic arthritis, and ankylosing spondylitis. It has demonstrated beneficial effects in alleviating pruritus, pain, and scaling [135,137,138,139]. Moreover, it demonstrates good efficacy in the treatment of extensive scalp psoriasis, improving patient outcomes and quality of life [140].

Compared to other biologics, such as ustekinumab and etanercept, secukinumab, with a comparable safety profile, provides better reduction in disease symptoms and greater improvement in quality of life [141]. Initiating therapy at an early stage of the disease may result in greater treatment benefits [142].

Additionally, the clinical efficacy of secukinumab was assessed in clinical trials involving patients with moderate-to-severe hidradenitis suppurativa. The results demonstrated rapid clinical improvement and a sustained therapeutic response [143].

##### Ixekizumab

Ixekizumab is a humanised, high-affinity IgG4κ monoclonal antibody that selectively binds and neutralises interleukin-17A (IL-17A) homodimers and IL-17A/F heterodimers, thus preventing their interaction with the IL-17 receptor. Its efficacy and safety in treating patients with moderate to severe plaque psoriasis or active psoriatic arthritis, particularly those who have had inadequate response to prior tumour necrosis factor inhibitor therapy, have been confirmed in clinical trials [135,144,145,146,147,148,149].

This drug is characterised by a rapid and lasting therapeutic response, with a significant clinical effect including notable reduction in pruritus severity and other psoriasis symptoms, as well as a clear improvement in patients’ quality of life. Another key advantage is its faster onset of therapeutic effect compared to IL-23 inhibitors [147,150,151].

##### Bimekizumab

Bimekizumab is a humanised IgG1 monoclonal antibody that potently and selectively neutralises the biological functions of both IL-17A and IL-17F, thus blocking their interaction with the IL-17 receptor. Although IL-17A exhibits stronger proinflammatory activity, IL-17F expression is higher in psoriatic skin. These cytokines can also form IL-17A/F heterodimers. Simultaneous blockade of both cytokines, which are key mediators of the Th17-dependent inflammatory axis, may lead to more effective suppression of inflammation than neutralising IL-17A alone in patients with moderate to severe psoriasis [152,153,154].

Clinical trials have shown that treatment with bimekizumab resulted in a higher percentage of patients achieving complete clearance of skin lesions (Psoriasis Area and Severity Index, PASI 100) compared with secukinumab. Additionally, a faster onset of action and greater efficacy were reported in comparison with adalimumab and ustekinumab, all while maintaining a similar safety profile. However, bimekizumab was associated with a higher incidence of oral candidiasis, which was typically mild to moderate in severity [153,154,155].

#### 5.1.3. Inhibitors of IL-23 (Th17 Axis)

Advances in understanding the pathogenesis of chronic inflammatory skin diseases, such as psoriasis, and their systemic consequences have been crucial for the development of biological therapies targeting the key modulators of the chronically activated inflammatory axes. Dysregulation of the Th17-dependent immune response is associated not only with chronic skin inflammation but also with comorbidities such as cardiovascular disease and metabolic syndrome [156,157].

In addition to IL-17, one of the key cytokines involved in the Th17-dependent immune pathway is interleukin-23 (IL-23). IL-23 inhibitors, such as ustekinumab, guselkumab, tildrakizumab, and risankizumab, represent safe and effective treatment options for moderate to severe psoriasis. Moreover, their beneficial clinical effects have been demonstrated in the long-term treatment of patients with psoriasis coexisting with metabolic syndrome, who may exhibit a reduced response to IL-17 inhibitors. However, this phenomenon remains a subject of ongoing clinical research [135,156,157,158,159].

##### Ustekinumab

Ustekinumab is a fully human monoclonal antibody that targets the p40 subunit shared by interleukins IL-12 and IL-23, thereby modulating both the Th1- and Th17-mediated immune pathways. It was approved by the U.S. Food and Drug Administration (FDA) in 2009 for the treatment of moderate to severe psoriasis [135,160,161,162].

A key advantage is its convenient dosing regimen, allowing for administration every 12 weeks while maintaining sustained clinical effectiveness, a favourable safety profile and high patient tolerability [160,163,164].

Currently, however, biological drugs that specifically block the p19 subunit of IL-23, such as guselkumab, risankizumab and tildrakizumab, enable a more precise therapeutic approach thanks to their selective mechanism of action [6,165,166,167].

##### Guselkumab

Guselkumab is a fully human monoclonal antibody that blocks the p19 subunit of interleukin-23 (IL-23). It was approved in 2017 by the U.S. Food and Drug Administration (FDA) and the European Medicines Agency (EMA) for the treatment of moderate to severe plaque psoriasis in adults. Additionally, it demonstrates efficacy in the treatment of active psoriatic arthritis [167,168,169,170,171].

Clinical trials have confirmed the high efficacy and good tolerability of guselkumab, including in patients whose previous treatment with other biologics—such as ustekinumab or adalimumab, a commonly used tumour necrosis factor (TNF) inhibitor—did not yield sufficient results. Furthermore, patients with persistent residual psoriasis lesions during ustekinumab therapy who switched to guselkumab showed improvements in skin clearance and quality of life [166,167,168,172].

Early initiation of guselkumab therapy after psoriasis diagnosis may contribute to achieving a better clinical response and a faster therapeutic effect [173].

##### Risankizumab

Risankizumab is a high-affinity humanised IgG1 monoclonal antibody that specifically binds to the p19 subunit of interleukin-23, selectively inhibiting this cytokine, which is key to the Th17-dependent immune pathway. Its high efficacy in the treatment of moderate to severe psoriasis has been confirmed in clinical trials [135,165,174,175,176].

This drug can be safely used in long-term psoriasis therapy, is well tolerated and demonstrates a more favourable clinical effect in terms of skin clearance from psoriatic lesions compared to ustekinumab and adalimumab [174,175,176,177,178,179].

##### Tildrakizumab

Tildrakizumab is a high-affinity humanised IgG1κ monoclonal antibody directed against the p19 subunit of interleukin-23. It is approved for the treatment of moderate to severe plaque psoriasis and has demonstrated good efficacy in this indication [180,181].

Its therapeutic effect, good safety profile, and positive impact on improving patients’ health-related quality of life (HRQoL) have been well documented in clinical trials [181,182,183,184].

The long-term efficacy of tildrakizumab has been confirmed in patients with psoriasis both with and without coexisting metabolic syndrome [185,186].

#### 5.1.4. Inhibitors of TNF-α

TNF-α is a proinflammatory cytokine that plays a significant role in the pathogenesis of chronic inflammatory skin diseases, including psoriasis. It exacerbates inflammation and facilitates the migration and infiltration of inflammatory cells into affected skin by inducing adhesion molecules on vascular endothelial cells. Once in the dermis, TNF-α stimulates keratinocytes to produce other inflammatory mediators, leading to hyperkeratinization and activation of dermal macrophages and dendritic cells [6,187].

Monoclonal antibodies directed against tumour necrosis factor alpha (TNF-α) include infliximab (a chimeric human-mouse monoclonal antibody) and adalimumab (a fully human antibody with low immunogenicity). Both bind TNF-α, thus preventing its interaction with TNF receptors. Etanercept, in contrast, is a receptor fusion protein consisting of two human TNF-α receptors fused to the Fc fragment of a human antibody, which similarly inhibits TNF-α signalling, thereby preventing its proinflammatory effects. They are effective and well tolerated, providing symptom relief, improved skin condition, and enhanced quality of life in patients with psoriasis [39,135,187,188,189].

TNF-α inhibitors are approved by the U.S. Food and Drug Administration (FDA) for the treatment of moderate to severe psoriasis, and their efficacy has also been demonstrated in psoriatic arthritis therapy. Additionally, studies suggest that their use may reduce the risk of cardiovascular events in patients with chronic inflammatory diseases. Among TNF-α inhibitors, infliximab is the most effective drug for treating psoriasis, followed by adalimumab, while etanercept exhibits the weakest therapeutic effect in this indication [187,190].

Currently, however, TNF-α inhibitors demonstrate lower therapeutic efficacy compared to newer, more selective IL-17 and IL-23 inhibitors, which is why they are increasingly being replaced in the treatment of chronic inflammatory skin diseases [191]. Table 2. provides an overview of the biologic agents and their mechanisms of immune modulation presented in this section.

### 5.2. Potential Effects on Ageing-Related Mechanisms

Until recently, chronic inflammatory skin diseases such as psoriasis, atopic dermatitis (AD), and hidradenitis suppurativa (HS) were considered to be conditions limited to the skin. However, increasing evidence now points to their systemic consequences, resulting from persistent immune activation and elevated levels of proinflammatory cytokines in the blood.

One of the most important consequences of this condition is the development of chronic, low-grade systemic inflammation, which accelerates biological ageing. This persistent inflammatory state contributes to endothelial dysfunction, oxidative stress, immunosenescence, and altered expression of ageing-related genes and proteins, collectively reflecting inflammageing [81,192,193,194,195,196,197,198,199,200].

The introduction of targeted biologics that selectively act on key inflammatory pathways (IL-4/13, IL-17A/F, IL-23, TNF-α) has opened new therapeutic possibilities for patients whose previous treatments were insufficient. Moreover, the discovery of these novel therapeutic methods has enabled exploration into whether modulation of the immune system can affect the molecular and vascular mechanisms of ageing. This represents a potentially groundbreaking direction in the treatment of chronic inflammatory skin diseases and may pave the way to addressing not only cutaneous symptoms but also the systemic health consequences of persistent inflammation.

Targeted immunomodulation and the ability to modify disease progression and counteract inflammageing may have significant benefits not only for skin health but also for the health of other tissues and organs affected by inflammation. This approach could lead to improved overall health and quality of life for patients, as well as reduce the risk of developing comorbidities and slow down biological ageing processes [187,201,202,203,204].

#### 5.2.1. Cytokine Suppression

The pathogenesis of chronic inflammatory skin diseases results from the dysregulation of proinflammatory cytokines that stimulate and activate immune cells. Persistent activation of inflammatory pathways, particularly the Th2 and Th17 axes, leads to elevated cytokine levels and chronic, low-grade systemic inflammation (inflammageing), thereby creating a self-perpetuating vicious cycle. Targeted biological therapy blocking key cytokines of the Th2 and Th17 axes, IL-4, IL-13, IL-17A/F, IL-23, and TNF-α, can effectively reduce inflammation and limit excessive immune system activation, potentially counteracting inflammageing [63,205,206,207,208,209]

A study by Florian TL et al. analysed the relationship between levels of TNF-α, IL-23, IL-17A, and IL-17F and the duration and severity of clinical symptoms in patients with psoriasis. The impact of biological therapy with ixekizumab, secukinumab, guselkumab, or adalimumab on reducing these cytokine levels was also evaluated. Disease severity was assessed using psoriasis severity indices such as the Psoriasis Area Severity Index (PASI) and the Dermatology Life Quality Index (DLQI). After three months of therapy, a reduction in inflammatory cytokine levels and improvements in PASI and DLQI scores were observed. These findings confirm that targeted cytokine blockade can effectively reduce chronic inflammation, leading to clinical improvement in patients [207].

Limiting inflammageing is particularly important in the context of chronic inflammatory dermatoses, as many comorbidities arise from immunological mechanisms driven by an excess of proinflammatory cytokines [210].

Targeted inhibition of IL-4, IL-13, IL-17A/F, IL-23, and TNF-α reduces systemic inflammation, potentially preventing its systemic complications by effectively lowering the risk of diseases associated with accelerated biological ageing, which is driven by excessive, chronic activation of inflammatory pathways.

For example, psoriasis, besides affecting the skin, often coexists with cardiovascular diseases, atherosclerosis, metabolic syndrome, type 2 diabetes, obesity, psoriatic arthritis (PsA), inflammatory bowel disease, and non-alcoholic fatty liver disease (NAFLD), all of which occur more frequently than in the general population [210,211,212]. The systemic nature of atopic dermatitis also leads to an increased risk of cardiovascular complications [81,213]. Moreover, the risk of ischemic stroke increases with the severity of atopic dermatitis [214].

The skin, as the body’s largest organ, serves as an immunological barrier but is also a significant source of proinflammatory cytokines that enter the bloodstream. Prolonged elevated levels of these cytokines, characteristic of inflammatory diseases, lead to premature immunosenescence of immune cells, resulting in immune system weakening and an increased risk of other diseases.

Although the mechanisms underlying inflammageing are not yet fully understood, biological therapy targeting proinflammatory cytokines may represent a promising and potentially effective strategy for preventing and treating this phenomenon. However, further research is necessary to develop therapeutic approaches that can modulate these processes [210,215].

#### 5.2.2. Downregulation of Ageing Biomarkers

Chronic skin diseases contribute to inflammageing by sustaining immune cell activation and altering ageing-related biomarkers [40,64,77,216]. This promotes vascular ageing and damage, as evidenced by accelerated vascular wall remodelling and enhanced atherogenesis, a key risk factor for the development of cardiovascular diseases [217].

Commonly analysed biomarkers potentially associated with ageing include IL-6, hs-CRP, VCAM-1, ICAM-1, and IL-17A [79,218,219,220] Additionally, evidence indicates a significant role for soluble selectins, such as sP-selectin, in endothelial dysfunction and the development of atherosclerosis, which may indirectly link them to cardiovascular ageing [46].

##### IL-6 and hs-CRP

Elevated levels of interleukin-6 (IL-6) and high-sensitivity C-reactive protein (hs-CRP) serve as biomarkers reflecting immunosenescence and the presence of active, chronic, low-grade inflammation, known as inflammageing. Their elevated concentrations correlate with an increased risk of morbidity, mortality, and frailty. Serum IL-6 levels can be used as indicators of disease activity and severity, and a decrease following biological therapy enables the assessment of treatment response in patients with psoriasis [218,221,222].

Tocilizumab, a humanised monoclonal antibody against the IL-6 receptor, inhibits IL-6 signalling by preventing its binding to IL-6R. Targeting the IL-6 pathway with biologic therapy has emerged as an innovative therapeutic approach in the treatment of rheumatic diseases such as rheumatoid arthritis. Other drugs targeting IL-6 include sirukumab, olokizumab, and clazakizumab [223].

Hs-CRP is an acute-phase protein and an independent, well-documented risk factor for atherosclerosis. A study conducted by Niknezhad N et al. demonstrated elevated hs-CRP levels in patients with psoriasis, suggesting that anti-inflammatory therapy, commonly used in chronic inflammatory diseases, could potentially reduce the risk of cardiovascular disease [224].

Reducing IL-6 and hs-CRP levels following biological therapy may therefore exert a beneficial effect on the inflammatory mechanisms of ageing, potentially delaying immunosenescence. Due to the lack of direct evidence, further research in this area is warranted.

##### VCAM-1 and ICAM-1

VCAM-1 and ICAM-1 are cell adhesion molecules that regulate immune and inflammatory responses by enabling leukocyte adhesion to the endothelium and migration into inflamed tissues. Their expression may be elevated in chronic inflammatory skin diseases, such as psoriasis [43,225].

Chronic activation of the inflammatory pathways and dysregulated, elevated cytokine levels lead to endothelial dysfunction and impaired vascular dilatation. The endothelium then acquires a proinflammatory phenotype, resulting in increased expression of VCAM-1 and ICAM-1. These molecules serve as markers of endothelial dysfunction and vascular damage, which represent key steps in the initiation and progression of atherosclerosis. Additionally, they may function as diagnostic markers for assessing cardiovascular risk in patients with chronic inflammatory diseases [43,226,227]

With age and biological ageing, serum levels of VCAM-1 and ICAM-1 increase, a change associated with the process of inflammageing [220].

TNF-α inhibitors, such as adalimumab, may improve endothelial function impaired in inflammatory dermatoses. In a study by Zdanowska et al., serum VCAM-1 levels in patients with plaque psoriasis decreased significantly after twelve weeks of adalimumab therapy, suggesting a correlation between treatment and reduced plasma levels of cell adhesion molecules.

Targeting biological therapy to reduce VCAM-1 and ICAM-1 levels may play a key role in preventing complications in individuals at high risk of developing cardiovascular disease [228].

##### IL-17A

IL-17A, a key mediator of the Th17-dependent inflammatory axis, plays a significant role in vascular ageing by promoting oxidative stress and premature endothelial cell senescence. These processes contribute to endotheliopathy, vascular damage, and accelerated atherosclerosis [18,79,80].

The accumulation of senescent endothelial cells promotes increased vascular stiffness, chronic inflammation, and vascular dysfunction characterised by impaired vascular tone, leading to accelerated vascular ageing [17].

Secukinumab and ixekizumab (anti-IL-17A antibodies), as well as bimekizumab (anti-IL-17A/F antibody), are effective drugs used in the biological therapy of psoriasis [229].

Krueger JG et al. conducted a randomised, double-blind, placebo-controlled clinical trial in patients with psoriasis to evaluate the efficacy of secukinumab, a fully human monoclonal antibody that selectively inhibits IL-17A. The results demonstrated that biologic therapy targeting IL-17A effectively reduced its levels, which correlated with clinical improvement [137].

Although direct research on the potential impact of biological therapies used in chronic inflammatory skin diseases on ageing mechanisms is currently limited, persistent inflammation may induce changes in ageing markers [64]. Consequently, targeted biological therapy that reduces IL-17A levels, a key factor in vascular ageing processes, could potentially slow these processes. However, these mechanisms are complex and multifactorial, warranting further clinical investigation

##### Soluble P-selectin

In chronic inflammatory skin diseases, such as moderate to severe psoriasis and atopic dermatitis, enhanced platelet activation is observed. In addition to their roles in hemostasis and thrombosis, platelets amplify inflammation and act as important modulators of the immune response [230,231,232,233].

A study by Tamagawa-Mineoka et al. demonstrated elevated plasma levels of soluble P-selectin in patients with psoriasis compared to healthy controls. Clinical improvement was associated with a reduction in its levels, suggesting its potential use as a marker of treatment response [231].

Long-term biological therapy with anti-TNF-α and anti-IL-12/23 drugs, such as infliximab, adalimumab, etanercept, and ustekinumab, can reduce P-selectin levels. This reduction lowers elevated platelet activation, a well-established risk factor for cardiovascular disease (CVD), the incidence of which increases with chronic inflammatory dermatoses. Therefore, P-selectin levels may potentially serve as a biomarker for assessing treatment efficacy in these conditions [230,232].

Although direct evidence supporting the role of P-selectin as a biomarker of ageing is currently lacking, its involvement in the pathophysiology of inflammation and endothelial dysfunction, key mechanisms involved in vascular ageing and inflammageing, as well as its elevated levels in chronic skin diseases, suggest a potential role in these processes. This highlights the need for further research, as targeted biological therapy could potentially prevent the systemic consequences of premature vascular ageing.

#### 5.2.3. Reduction in Oxidative Stress

The skin, as an active immunological barrier, is a major source of free radicals, whose excess damages DNA, cellular proteins, and lipids, ultimately resulting in tissue injury. This disrupted redox balance contributes to increased IL-17 production [234,235].

Oxidative stress plays a key role in the pathogenesis of inflammatory diseases such as psoriasis and atopic dermatitis. Redox imbalance and chronic inflammation mutually amplify each other, creating a vicious cycle. The intensified inflammatory response further enhances ROS production while impairing antioxidant capacity. These phenomena are observed both in the skin and systemically, in the plasma and blood, emphasising the systemic nature of these disorders [234,235,236,237].

Excess ROS promotes inflammageing, accelerating the development of diseases associated with biological ageing [196]. Additionally, oxidative stress enhances endothelial cell senescence, thereby promoting vascular ageing and chronic inflammation [17].

Overexpression of IL-17A in the skin has been shown to induce systemic endothelial dysfunction, vascular oxidative stress, hypertension, and increased mortality [200]. In the serum of patients with psoriasis, higher levels of oxidant markers—including advanced glycation end products (AGEs) and advanced oxidation protein products (AOPPs)—and lower levels of antioxidant enzymes—including lecithin-cholesterol acyltransferase (LCAT), paraoxonase-1 (PON1), and ferric-reducing ability of plasma (FRAP)—are observed, confirming a link between chronic inflammation and oxidative damage [238].

Biological drugs, such as IL-17A, IL-12/23, and TNF-α inhibitors, used in the treatment of chronic inflammatory dermatoses, exhibit potential antioxidant effects by reducing oxidative stress, limiting persistent inflammation driven by free radicals, and modulating the immune response [236,239,240,241,242].

For example, in a clinical study conducted by Bacchetti et al., treatment with etanercept was associated with reduced lipid peroxidation and improved antioxidant parameters, correlating with clinical improvement [240]. Reductions in oxidative stress and enhanced total antioxidant capacity were also observed in patients treated with secukinumab [241].

Interactions between biological therapies and immune and redox-related signalling pathways require further investigation, as the potential antioxidant effects of these drugs may prove effective in treating refractory cases of chronic inflammatory skin diseases [236].

#### 5.2.4. Improved Endothelial Function

Chronic, systemic inflammation characteristic of inflammatory skin diseases leads to endothelial dysfunction, a key factor in the development of atherosclerosis and cardiovascular disease. Endothelial function can be assessed using flow-mediated dilation (FMD), which reflects the ability of vessels to dilate in response to increased blood flow. Optimal control of inflammation in psoriasis, achieved through inhibition of the IL-23/Th17 inflammatory axis, may help prevent these complications. Biological therapy targeting IL-17A, which contributes to accelerated endothelial cell ageing, improves vascular function and helps prevent ageing-related complications [79,201,203].

The CARIMA study, a 52-week, randomised, double-blind, placebo-controlled exploratory trial, assessed endothelial function in patients with moderate-to-severe plaque psoriasis treated with secukinumab. A significant increase in FMD from baseline was observed [203].

Systemic inflammation can also affect cardiovascular function by promoting inflammation of the arterial walls. Intima-media thickness (IMT) is a marker of early atherosclerosis and can be measured by ultrasound examination of large arteries such as the common carotid, brachial, and common femoral arteries. Patients with severe psoriasis have been shown to exhibit increased carotid IMT (cIMT) compared to healthy controls [243,244,245].

A prospective study by Martinez-Lopez et al. demonstrated that biologic therapy, particularly with anti-IL-12/23 antibodies, can reduce carotid intima-media thickness (cIMT) in patients with moderate to severe psoriasis. This suggests a beneficial effect of reducing chronic systemic inflammation on vascular health and its related outcomes [246].

Increased levels of soluble markers of endothelial activation, such as vascular cell adhesion molecule-1 (VCAM-1) and E-selectin, are also observed in patients with chronic systemic inflammatory diseases. Biological therapies, including TNF-α inhibitors, may exert a positive effect by improving endothelial cell function [228,247].

Another biomarker of vascular endothelial damage is the level of circulating endothelial cells (CECs), which has been found to be elevated in patients with psoriasis. After six months of treatment with etanercept, a TNF-α inhibitor, a significant decrease in CEC levels was observed, correlating with clinical improvement. The reduction in CEC levels following therapy confirms that targeted biological treatment can improve endothelial function, reducing the risk of complications arising from its dysfunction [248].

#### 5.2.5. Preservation of Mitochondrial Function

Mitochondria are currently recognised as key determinants of ageing and the exacerbation of inflammation in senescent cells. Biological therapies and their potential impact on mitochondrial dysfunction may therefore contribute to delaying the ageing process.

Mitochondria play a central role in proinflammatory signalling, while also being susceptible to proinflammatory mediators that can alter their function. This interrelationship exacerbates mitochondrial oxidative stress, creating a vicious cycle of inflammation. Ageing, dysfunctional, or damaged mitochondria may promote inflammation through continuous stimulation of the immune system.

In chronic inflammatory skin diseases such as psoriasis, mitochondrial dysfunction may contribute to disease pathogenesis through three mechanisms: (i) oxidative stress and disrupted lipid metabolism, (ii) mitochondrial alterations in keratinocytes following pharmacological treatment, and (iii) elevated serum levels of mitochondrial DNA (mtDNA), which trigger inflammation [249,250,251].

A study by Di Vincenzo et al. demonstrated that adalimumab, a drug that inhibits the TNF-α inflammatory cascade, can partially improve mitochondrial function. This suggests that mitochondria may serve as a potential therapeutic target for TNF-α inhibitors. However, adalimumab did not prevent mitochondrial dysfunction under conditions of additional oxidative or inflammatory stress, which could lead to symptom recurrence after treatment [249].

Given the role of mitochondria in inflammation and ageing, biological therapies targeting their function may modulate inflammageing processes in chronic inflammatory skin diseases. However, further clinical research is needed to confirm this.

#### 5.2.6. Modulation of Longevity-Associated Factors

##### SIRT1

Sirtuin 1 (SIRT1), associated with longevity, plays a crucial role in regulating ageing and inflammation. It is involved in key biological processes such as transcriptional regulation, chromosomal stability, DNA repair, lipid and carbohydrate metabolism, and oxidative stress management. SIRT1 modulates inflammation-related signalling pathways and suppresses the production of reactive oxygen species (ROS) in mitochondria, thereby reducing oxidative stress, mitochondrial DNA mutations, and mitochondrial damage [252,253,254].

The diverse functions of sirtuin 1 suggest its beneficial role in delaying skin ageing [255]. Studies have shown that SIRT1 expression is significantly decreased in skin affected by psoriatic lesions, higher in perilesional areas, and highest in the skin of healthy individuals. SIRT1 may limit TNF-α activity by deacetylating NF-κB, leading to inhibition of TNF-α transcription and reduced expression of TNF-α-induced proinflammatory cytokines. Conversely, TNF-α, a key inflammatory factor in psoriasis, may downregulate SIRT1 expression.

In the future, SIRT1 activators may represent a novel, targeted therapeutic strategy for chronic inflammatory skin diseases, potentially preventing skin ageing and slowing vascular ageing [252,256,257].

It is also possible that biologic drugs, such as TNF-α inhibitors, may prevent the downregulation of SIRT1, which in turn could reduce inflammation in chronic skin diseases and potentially delay the ageing process. However, these mechanisms require further investigation and validation.

##### Klotho

Klotho may exert protective effects against diseases associated with biological ageing by regulating multiple pathways involved in these processes. Klotho exhibits antioxidant and vasoprotective properties, protects stem cells, and inhibits inflammatory responses through the modulation of signalling pathways. Its levels decrease as systemic inflammation increases, as measured by the systemic immune-inflammation index (SII). Both Klotho expression and circulating concentrations decline with age, and this reduction may serve as an early marker of premature vascular ageing and atherosclerosis [258,259,260,261].

There are no direct studies assessing the role of Klotho in the context of chronic inflammatory skin diseases. However, its mechanisms of action as an anti-ageing factor, along with its ability to inhibit inflammageing characteristic of these conditions, suggest that modulation of Klotho levels, potentially through the use of biologic therapies, could effectively prevent systemic complications associated with accelerated biological ageing in this patient population.

#### 5.2.7. Disease-Modifying Potential

Long-term biologic therapy can modulate the natural course of chronic inflammatory skin diseases. IL-23 inhibitors (e.g., risankizumab) and IL-4/13 blockers (e.g., dupilumab) have been shown to exert effects beyond symptom control, potentially reversing molecular damage and suppressing the chronic immune response.

A study by Visvanathan et al. compared the molecular and cellular responses to risankizumab and ustekinumab in patients with psoriasis. Histological improvement, along with decreases in protein and RNA biomarkers associated with psoriasis, were observed after both treatments. Risankizumab induced more pronounced changes in the molecular and histopathological profiles of psoriatic skin lesions compared with ustekinumab. Furthermore, it more strongly modulated gene expression, significantly reducing the expression of genes related to keratinocytes, epidermal cells, and monocytes, which may explain its superior control of skin inflammation [262].

Hamilton et al. examined the effect of dupilumab on the molecular profile of skin in patients with moderate to severe atopic dermatitis. Transcriptomic analysis revealed a significant, dose-dependent reduction in the expression of genes active in the lesions, including those related to hyperplasia, T lymphocytes, and dendritic cells. Inhibition of chemokines associated with the Th2-dependent inflammatory axis was also observed. These molecular changes correlated with clinical improvement [121].

Targeted pharmacotherapy can therefore modify the course of chronic inflammatory skin diseases by normalising gene expression and the phenotype of immune cells [263].

Research findings indicate that biologic therapies have a significant impact on the molecular mechanisms underlying these diseases, potentially enabling the use of these pathways in other therapeutic contexts, such as limiting systemic complications and ageing processes. However, further studies are needed in this area.

## 6. Translational Relevance: Perspectives and Future Directions

In recent years, knowledge about chronic inflammatory skin diseases, such as psoriasis and atopic dermatitis (AD), has advanced significantly. This has enabled the discovery of their systemic nature and their connection to the process of inflammageing, which accelerates the biological ageing of the body.

The prolonged release of proinflammatory cytokines from the epidermis into the bloodstream promotes systemic inflammation, contributing to accelerated molecular and vascular ageing. These mechanisms represent a key factor in the development of inflammageing and its associated consequences, including the accelerated progression of age-related diseases [60,198,264].

Biologic drugs have the potential to slow inflammation-induced molecular ageing by precisely blocking selected cytokines. Understanding the roles of various cytokines and immune pathways has enabled the development of promising new and effective therapeutic strategies. A key factor in this translational revolution has been the integration of pathogenesis research with the identification of biomarkers for therapeutic response [264].

Potential long-term benefits of targeted biological therapies include improved vascular function and reduced risk of comorbidities, including cardiovascular diseases. These drugs may also reduce oxidative stress, thereby slowing the ageing of the immune system, and restoring molecular anti-ageing pathways involving SIRT1 and Klotho. Additionally, disease modification—achieving durable symptom control, reducing relapse frequency, and potentially reversing molecular damage—represents a promising therapeutic prospect.

Further longitudinal and mechanistic studies are needed to confirm the role of biologics as disease-modifying therapies and as agents preventing ageing in both the skin and the circulatory system.

## 7. Conclusions

Chronic inflammatory skin diseases such as atopic dermatitis, psoriasis or hidradenitis suppurativa should be viewed not only as skin-limited conditions but also as systemic diseases, serving as valuable clinical models for studying inflammageing, a chronic, low-grade inflammation that accelerates the biological ageing process.

Persistent, active skin inflammation can act as a trigger for premature ageing at the molecular and endothelial levels. Recent studies indicate a link between these conditions and endothelial dysfunction, oxidative stress, mitochondrial dysregulation, and altered expression of key longevity regulators such as Klotho and SIRT1.

Systemic ageing is driven by inflammatory Th17 and Th2 pathways and their associated cytokines, including IL-17, IL-23, IL-6, IL-4, IL-13, and IL-31. Immune cell senescence and vascular biomarkers, such as VCAM-1 and ICAM-1, which are markers of inflammageing, also play a significant role.

Biological treatments targeting key modulators of these inflammatory axes—IL-4/IL-13, IL-17, and IL-23—have the potential to modulate inflammation-driven ageing processes. The development of targeted therapies opens new therapeutic avenues, offering the opportunity to address not only skin symptoms but also systemic consequences of chronic inflammation. This holds promise for improving patients’ overall health and quality of life. Further clinical and experimental studies are needed to evaluate the potential of biological therapies in mitigating systemic ageing associated with these inflammatory diseases.

## Figures and Tables

**Figure 1 cells-14-01442-f001:**
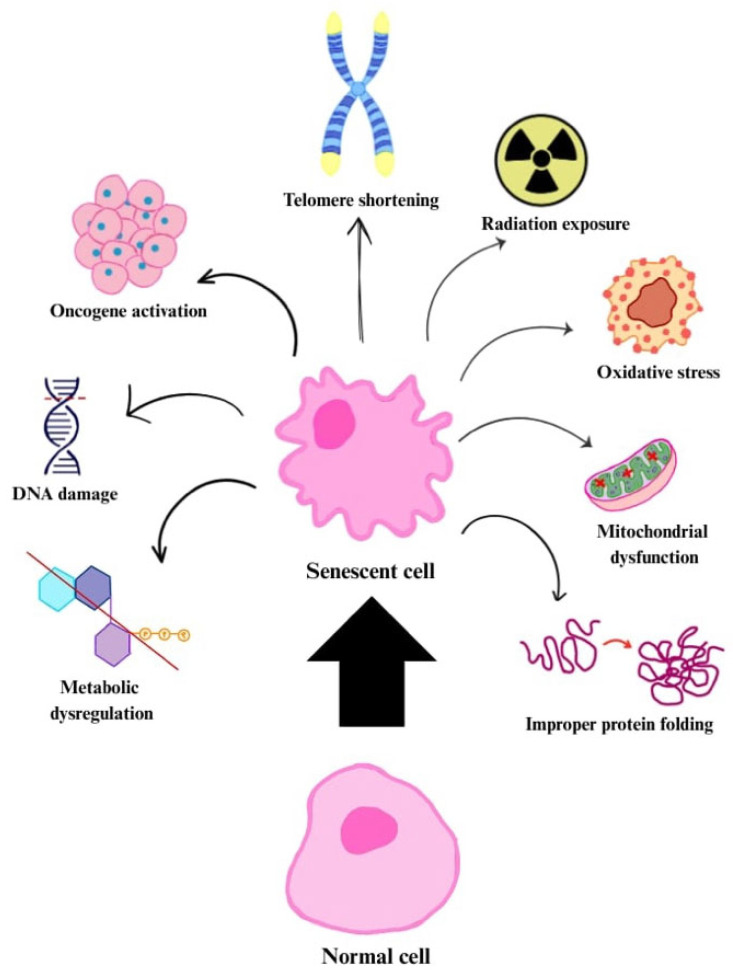
Inducers of cellular senescence.

**Figure 2 cells-14-01442-f002:**
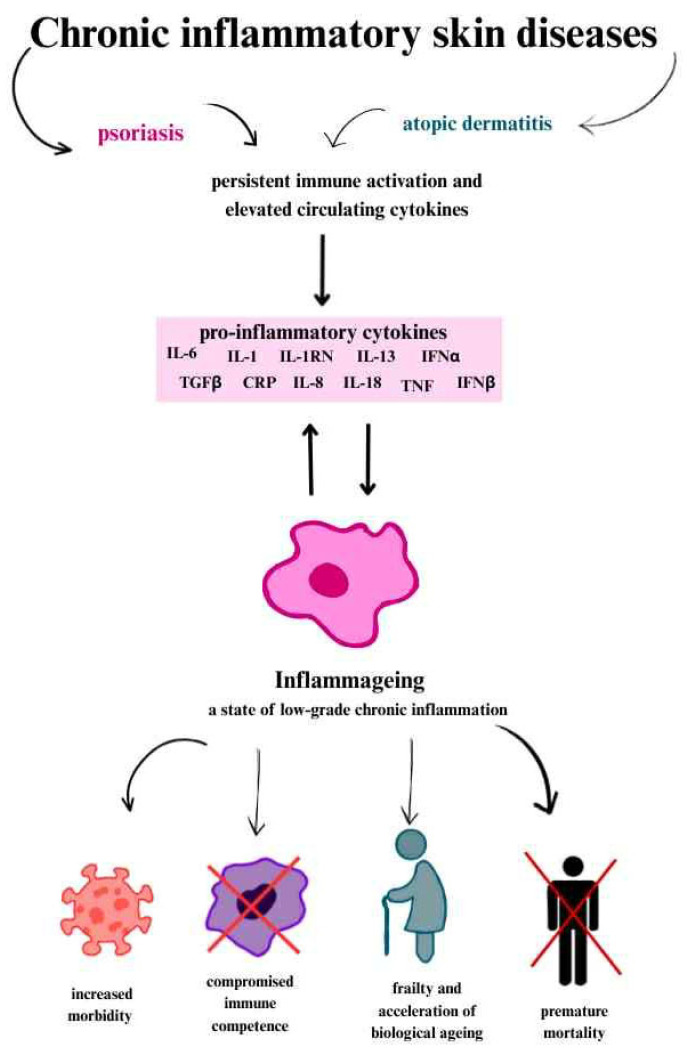
Chronic inflammation and cytokine-driven consequences in skin diseases.

**Figure 3 cells-14-01442-f003:**
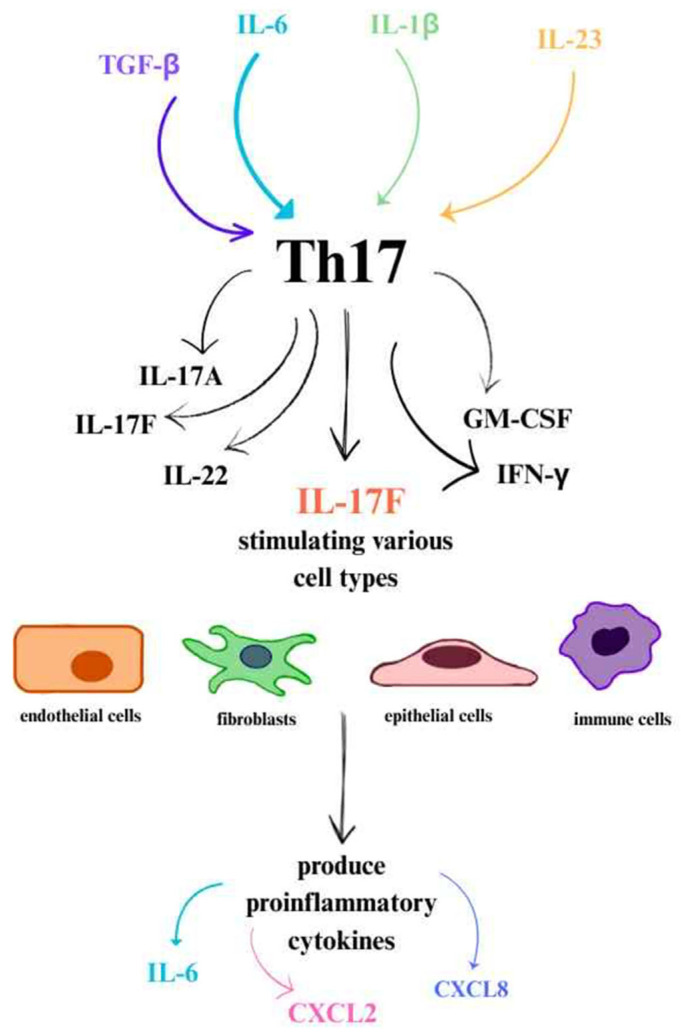
Key cytokines driving pathogenic Th17 cell responses.

**Table 1 cells-14-01442-t001:** Contribution of major cytokines to ageing-associated processes and inflammageing.

Cytokine	Systemic Chronic Inflammation	Oxidative Stress	Endothelial Activation	SASP	Barrier Dysfunction	Neuro-Inflammation	References
Th17							
IL-17	✓	✓	✓	✓			[31,32,33,34]
IL-23	✓			✓			[31,32]
Th2							
IL-4	✓				✓		[35,36]
IL-13	✓				✓		[35,36]
IL-31	✓					✓	[37]
Others							
IL-6	✓			✓			[26,27]
IL-8	✓			✓			[27]
TNF-α	✓		✓	✓			[27,38,39]

**Table 2 cells-14-01442-t002:** Biologic agents: molecular targets and mechanisms of immune modulation.

Biologic Agent	Target	Mechanism of Action	References
Inhibitors of IL-4/IL-13 (Th2 axis)			
Dupilumab	IL-4Rα	Blocking the signalling of IL-4 and IL-13	[117]
Tralokinumab	IL-13	Preventing receptor interaction and downstream signalling	[127]
Lebrikizumab	IL-13	Preventing the formation of the IL-4Rα/IL-13Rα1 heterodimer receptor signalling complex	[131,132,133]
Inhibitors of IL-17/IL-17F (Th17 axis)			
Secukinumab	IL-17A	Preventing interactions between IL-17A and its receptor and downstream signalling	[140,141,142]
Ixekizumab	IL-17A; IL-17A/F	Preventing interactions between IL-17 and IL-17 receptors	[147,148,149,150,151,152]
Bimekizumab	IL-17A; IL-17F	Blocking the interaction of IL-17 and IL-17F with the IL-17 receptor	[155,156,157]
Inhibitors of IL-23 (Th17 axis)			
Ustekinumab	p40 subunit shared by IL-12 and IL-23	Modulating of the Th1 and Th17-mediated immune pathways	[163,164,165]
Guselkumab	p19 subunit of IL-23	Blocking the IL-23, leading to suppression of the Th-17 dependent immune pathway	[170,171,172,173,174]
Risankizumab	p19 subunit of IL-23	Blocking the IL-23, leading to suppression of the Th-17 dependent immune pathway	[177,178,179]
Tildrakizumab	p19 subunit of IL-23	Blocking the IL-23, leading to suppression of the Th-17 dependent immune pathway	[183,184]
Inhibitors of TNF-α			
Infliximab	TNF-α	Preventing interaction between TNF-α and its receptors	[138,192]
Adalimumab	TNF-α	Preventing interaction between TNF-α and its receptors	[138,193]
Etanercept	TNF-α	Receptor fusion protein inhibiting TNF-α signalling	[138,194]

## Data Availability

Not applicable.

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
