# Peer review of "Inflammation-Driven Molecular Ageing in Chronic Inflammatory Skin Diseases: Is There a Role for Biologic Therapies?"

_cells, 2025, doi:10.3390/cells14181442_

Round 1
Reviewer 1 Report
Comments and Suggestions for Authors
While the topic presented by Andrzejczak et al, of aging and immune responses in skin diseases, is interesting, several concerns need to be addressed to enhance the quality and clarity of the review. In particular, there are issues related to clarity, insufficient references, poor organization, and a misalignment between the title and content.
- The authors should include key references in the introduction to support the background and rationale.
- Several important features of aging and cellular senescence are missing. Notably, thymic involution and hematopoietic stem cell dysfunction are established hallmarks that should be included for a more comprehensive overview.
- The table lacks references regarding the role of cytokines in aging and related processes.
- While the authors discuss aging in the context of skin diseases, the review lacks sufficient explanation and differentiation of the diseases mentioned. For instance, the discussion of Th17 in atherosclerosis and vascular aging seems disconnected from the focus on skin diseases and appears out of context.
- The manuscript appears to be more of a review on immune mechanisms related to skin diseases rather than a focused discussion on the process of inflammaging. The title and structure should align with the actual content, or the content should be revised to match the stated scope.
- There is significant repetition throughout the text, which affects readability. Additionally, the absence of visual aids (e.g., figures, diagrams) makes it difficult to follow complex information. Including summary figures would greatly enhance comprehension and overall engagement.
Author Response
We sincerely thank the Reviewer for the careful and detailed evaluation of our manuscript. We truly appreciate the constructive criticism and recognize that the concerns raised have significantly contributed to improving the quality, organization, and clarity of our review. Below we provide a point-by-point response to the Reviewer’s comments.
Comment 1:
The authors should include key references in the introduction to support the background and rationale.
Response:
We thank the Reviewer for this important comment. In response, we carefully re-analyzed the Introduction and strengthened the scientific background. We included additional references covering systemic features of immunosenescence, inflammaging, and its relevance to skin diseases [1–7]. These additions clarify the rationale of the review and better position our work within the context of the existing literature.
Comment 2:
Several important features of aging and cellular senescence are missing. Notably, thymic involution and hematopoietic stem cell dysfunction are established hallmarks that should be included for a more comprehensive overview.
Response:
We appreciate the Reviewer’s valuable suggestion. In response, we expanded the scope of Section 2 and introduced a new subsection (2.1, Systemic Features of Immunosenescence), in which we provide a detailed discussion of the hallmarks of immunosenescence, including thymic involution and hematopoietic stem cell dysfunction, emphasizing their systemic consequences. We also added relevant references [8–13] to support this section. These additions enrich the manuscript, broaden its thematic coverage, and provide a more comprehensive overview of immunosenescence, thereby strengthening the biological foundations of our review.
Comment 3:
The table lacks references regarding the role of cytokines in aging and related processes.
Response:
We are grateful to the Reviewer for this insightful comment. In response, we have thoroughly revised Table 1. We included relevant references supporting the role of each cytokine in ageing-associated processes, and also updated the table title and caption to make them fully explicit. These revisions enhance the clarity, scientific rigor, and accessibility of the table, providing readers with a clear and informative overview of cytokine involvement in ageing-associated processes and inflammaging.
Comment 4:
While the authors discuss aging in the context of skin diseases, the review lacks sufficient explanation and differentiation of the diseases mentioned. For instance, the discussion of Th17 in atherosclerosis and vascular aging seems disconnected from the focus on skin diseases and appears out of context.
Response:
We thank the Reviewer for their constructive and valuable feedback. In response, we have removed Section 4.1.5 (The Impact of the IL-23/IL-17 Axis on Comorbidities and Atherosclerosis), which was excessively detailed and not directly relevant to skin diseases. This revision improves the focus and coherence of the manuscript, ensuring that the discussion of inflammaging is fully aligned with the dermatological context and provides a more streamlined and impactful narrative for the reader.
Comment 5:
The manuscript appears to be more of a review on immune mechanisms related to skin diseases rather than a focused discussion on the process of inflammaging. The title and structure should align with the actual content, or the content should be revised to match the stated scope.
Response:
We acknowledge the Reviewer’s insightful observation. We have carefully reviewed and refined the manuscript to ensure clarity, focus, and coherence, ensuring that the discussion of inflammaging is well integrated within the context of skin diseases.
Comment 6:
There is significant repetition throughout the text, which affects readability.
Response:
We fully agree with the Reviewer’s observation. In response, we carefully reviewed the manuscript and removed repetitive descriptions and definitions throughout the text. These revisions improve readability, clarity, and the scientific impact of the manuscript.
Comment 7:
Additionally, the absence of visual aids (e.g., figures, diagrams) makes it difficult to follow complex information. Including summary figures would greatly enhance comprehension and overall engagement.
Response:
We appreciate the Reviewer’s comment regarding the importance of visual aids for presenting complex information. In response, we have incorporated two new figures and one new table:
- Figure 2: Chronic inflammation and cytokine-driven consequences in skin diseases
- Figure 3: Key cytokines driving pathogenic Th17 cell responses
- Table 2: Biologic agents: molecular targets and mechanisms of immune modulation
These additions greatly enhance comprehension, facilitate understanding of complex mechanisms, and improve overall engagement. We fully agree that the inclusion of these visual aids enriches the manuscript and makes the content more accessible and informative.
Closing:
We are very grateful for the Reviewer’s thoughtful feedback, which has been invaluable in improving the manuscript’s clarity, coherence, and scientific depth. These revisions have strengthened the focus and organization of our review, and we hope the manuscript is now significantly improved.
Reviewer 2 Report
Comments and Suggestions for Authors
Overall, the review is well organized and clearly written. However, it would benefit from minor editing and revision taking into consideration the following points:
- It is recommended to add a schematic illustration for the mechanistic pathways of chronic inflammation and the role of biologic therapies in modulating aging. This would provide a clear visual explanation and enhance the readability.
- The introduction section outlined the aim of this review and introduced the concept of inflammaging. However, it remains brief for a narrative review. Moreover, no references were cited in this section. It is recommended to merge section 2.1 into the introduction to enhance the clarity of the concept of inflammaging and to strengthen the introduction section of this narrative review.
- Table 1 requires more clarification of the title and caption.
- Section 5; addition of a Table presenting the biologic therapies and their potential to modulate ageing pathways would strengthen the manuscript.
I appreciate the effort invested in this submission and encourage the authors to address these concerns. I hope this feedback is constructive for improvement of the manuscript.
Thank you for considering this review.
Author Response
We sincerely thank the Reviewer for the careful evaluation of our manuscript and for the constructive comments that helped us improve the quality and clarity of the paper. We have revised the manuscript accordingly. Below we provide a point-by-point response to each comment.
Comment 1:
It is recommended to add a schematic illustration for the mechanistic pathways of chronic inflammation and the role of biologic therapies in modulating aging. This would provide a clear visual explanation and enhance the readability.
Response:
We thank the Reviewer for this valuable suggestion. In response, we have created a new Table 1, which summarizes the mechanistic pathways linking chronic inflammation, cytokine-driven processes, and molecular ageing. We believe that this tabular presentation provides a clear and structured overview of the key mechanisms and enhances the readability of the manuscript.
Comment 2:
The introduction section outlined the aim of this review and introduced the concept of inflammaging. However, it remains brief for a narrative review. Moreover, no references were cited in this section. It is recommended to merge section 2.1 into the introduction to enhance the clarity of the concept of inflammaging and to strengthen the introduction section of this narrative review.
Response:
We agree with the Reviewer’s observation. To address this, we merged Section 2.1 (“Systemic Features of Immunosenescence”) with the Introduction. We also expanded the Introduction to provide a more comprehensive overview of inflammageing, and we added relevant references [1–5]. These changes provide a stronger background and clearer context for the aims of the review.
Comment 3:
Table 1 requires more clarification of the title and caption.
Response:
We have revised the title and caption of Table 1 to make them more explicit and reader-friendly. The updated caption now clearly states the role of major cytokines in ageing-associated processes and explains the meaning of the check marks.
Comment 4:
Section 5; addition of a Table presenting the biologic therapies and their potential to modulate ageing pathways would strengthen the manuscript.
Response:
We fully agree with the Reviewer’s recommendation. Therefore, we added a new Table 2, which presents biologic therapies, their molecular targets, mechanisms of immune modulation, and potential effects on ageing pathways. This addition provides a concise overview of currently available agents and strengthens the manuscript by facilitating comparison across therapeutic strategies.
Closing:
We are grateful for the Reviewer’s constructive comments, which have helped us improve both the structure and clarity of the manuscript.
Reviewer 3 Report
Comments and Suggestions for Authors
This is a very well-written review on a very complicated subject related to the role of inflammation in disease and aging. The authors discuss events involved in senescence, focus on Th17 and Th2 inflammatory events, and then discuss current biologics and their role in addressing inflammatory events that cause disease and aging. The inflammatory mediators and pathways involved are complicated and the average reader may find it difficult to follow all of the pathways and events.
My recommendation for making this paper more "readable" would be to include more graphics like the one you did for Figure 1. Creating a graphic that shows, for example, the events that activate Th17, the inflammatory mediators produced and the effect these have on the aging process would be very helpful for the reader.
My other recommendation concerns the discussion of biologics and their targets. For some of the biologics, it is clear from the manuscript how they work: e.g. Dupilumab binds the IL-4 receptor preventing IL-4 and IL-13 from signaling through this receptor. But for others the wording in the manuscript doesn't precisely identify how these biologics work. For example, Secukinumab is said to "inhibit IL-17A". Does this mean that it binds to IL-17A and prevents it from interacting with it's receptor or does it bind to the receptor thereby preventing Il-17 from binding. Of course, one can look up the biologics to determine precisely how they work, but it would be good to provide a clear statement of exactly how they work.
Other than these 2 suggestions, the manuscript is very good and will be useful as resource material.
Author Response
We sincerely thank the Reviewer for the very positive evaluation of our work and for the constructive comments that helped us to further improve the clarity and readability of the manuscript. Below we provide a point-by-point response.
Comment 1:
My recommendation for making this paper more "readable" would be to include more graphics like the one you did for Figure 1. Creating a graphic that shows, for example, the events that activate Th17, the inflammatory mediators produced and the effect these have on the aging process would be very helpful for the reader.
Response:
We thank the Reviewer for this excellent suggestion. In response, we expanded the graphical content of the manuscript by creating additional schematic illustrations. Specifically, we added a figure illustrating the Th17 axis, showing the key activating events, the inflammatory mediators produced, and their contribution to ageing processes. We believe this addition makes the review more accessible and easier to follow for the general readership.
Comment 2:
My other recommendation concerns the discussion of biologics and their targets. For some of the biologics, it is clear from the manuscript how they work: e.g. Dupilumab binds the IL-4 receptor preventing IL-4 and IL-13 from signaling through this receptor. But for others the wording in the manuscript doesn't precisely identify how these biologics work. For example, Secukinumab is said to "inhibit IL-17A". Does this mean that it binds to IL-17A and prevents it from interacting with it's receptor or does it bind to the receptor thereby preventing Il-17 from binding. Of course, one can look up the biologics to determine precisely how they work, but it would be good to provide a clear statement of exactly how they work.
Response:
We fully agree with the Reviewer’s observation. Accordingly, we revised Section 5 to provide more precise descriptions of the mechanisms of action of each biologic therapy. For example, we now specify that Secukinumab is a monoclonal antibody that binds directly to IL-17A, preventing its interaction with the IL-17 receptor. Similar clarifications were added for ixekizumab, bimekizumab, guselkumab, risankizumab, tildrakizumab, and other biologics. We also summarized this information in Table 2, which clearly presents each biologic, its molecular target, and the mechanism of immune modulation. These changes improve clarity and enhance the manuscript’s value as a reference.
Closing:
We are grateful for the Reviewer’s thoughtful comments, which helped us improve the clarity of our review and make it more useful as a resource for both specialists and general readers.